# Bispecific Antibodies in Cancer Immunotherapy: A Novel Response to an Old Question

**DOI:** 10.3390/pharmaceutics14061243

**Published:** 2022-06-11

**Authors:** Camila Ordóñez-Reyes, Juan Esteban Garcia-Robledo, Diego F. Chamorro, Andrés Mosquera, Liliana Sussmann, Alejandro Ruiz-Patiño, Oscar Arrieta, Lucia Zatarain-Barrón, Leonardo Rojas, Alessandro Russo, Diego de Miguel-Perez, Christian Rolfo, Andrés F. Cardona

**Affiliations:** 1Foundation for Clinical and Applied Cancer Research—FICMAC, Bogotá 110111, Colombia; camila.ordonez.reyes@gmail.com (C.O.-R.); garcia-robledoje@outlook.com (J.E.G.-R.); df.chamorro10@uniandes.edu.co (D.F.C.); andresfe.mosquerap@gmail.com (A.M.); alejandro.ruiz.pat@gmail.com (A.R.-P.); llrojas@colsanitas.com (L.R.); 2Molecular Oncology and Biology Systems Research Group (Fox-G), Universidad el Bosque, Bogotá 110121, Colombia; 3Division of Hematology/Oncology, Mayo Clinic, Phoenix, AZ 85054, USA; 4Department of Neurology, Fundación Universitaria de Ciencias de la Salud, Bogotá 111221, Colombia; lisussmann@hotmail.com; 5Thoracic Oncology Unit and Personalized Oncology Laboratory, National Cancer Institute (INCan), Mexico City 14080, Mexico; oscararrietaincan@gmail.com (O.A.); lucia.zatarain.barron@gmail.com (L.Z.-B.); 6Medical Oncology Unit, A.O. Papardo, 98158 Messina, Italy; ale.russo1986@gmail.com; 7Center for Thoracic Oncology, Tisch Cancer Institute and Icahn School of Medicine at Mount Sinai, New York, NY 10029, USA; diego.demiguelperez@mssm.edu (D.d.M.-P.); christian.rolfo@mssm.edu (C.R.); 8Direction of Research, Science and Education, Luis Carlos Sarmiento Angulo Cancer Treatment and Research Center (CTIC), Bogotá 110131, Colombia

**Keywords:** bispecific antibodies, immunotherapy, immune restoration, cancer therapy

## Abstract

Immunotherapy has redefined the treatment of cancer patients and it is constantly generating new advances and approaches. Among the multiple options of immunotherapy, bispecific antibodies (bsAbs) represent a novel thoughtful approach. These drugs integrate the action of the immune system in a strategy to redirect the activation of innate and adaptive immunity toward specific antigens and specific tumor locations. Here we discussed some basic aspects of the design and function of bsAbs, their main challenges and the state-of-the-art of these molecules in the treatment of hematological and solid malignancies and future perspectives.

## 1. Introduction

Regardless of efforts from the scientific community, options to treat cancer patients in advanced stages achieving complete response with low recurrence are limited. For that reason, the search of effective alternatives to treat cancer has increase in the last years. Currently, some alternatives that are effective in the treatment of other conditions are now being studied as an alternative to treat cancer patients. Monoclonal antibodies are known to have a positive impact on many conditions such as autoimmune disorders, cardiovascular, pulmonary, and even infectious diseases [1]. Even though monoclonal antibodies are usually specific to one epitope, genetic and cell engineering have allowed the biosynthesis of bispecific antibodies (bsAbs). BsAbs were first described by Nisonoff et al. over 60 years ago; however, they gained clinical relevance after the first approval by the Food and Drugs Administration (FDA) [2] of blinatumomab, a bsAb approved for the treatment of acute myeloid leukemia. Since then, these molecules have become an attractive choice to treat cancer, due to their efficacy and safety profile (Figure 1) [3]. The original concept of bsAbs was a molecule that can bind to two different epitopes [2].

The first application of bsAbs in cancer immunotherapy was focused on leading T cells toward tumor cells by the interaction between the extracellular subunit of CD3 on T cells and cancer-related antigens. The bsAbs ease the interaction of the major histocompatibility complex (MHC) with its cognate T-cell receptor (TCR) resulting in a proper T-cell priming and activation. Despite this, some adverse effects of these drugs such as cytokine release syndrome or liver toxicity and other limitations such as a short half-life have been reported. For that matter a vast quantity of clinical trials with these molecules is being conducted [4].

Nevertheless, bsAbs still represent a novel and effective approach to treat cancer patients because they target molecules expressed on the surface of cancer cells (tumor-associated antigens [TAAs]) and bind to specific receptors that are located on effector cells of the immune system (Figure 2) [5,6]. Furthermore, there have been other smart approaches for the use of bsAbs. Fournier et al. used the Newcastle Disease Virus to specifically infect cancer cells and make them express viral antigens such as hemagglutinin-neuraminidase and fusion molecules. By expressing these viral antigens, bsAbs can be engineered to engage immune effector cells to cancer cells, decreasing the risk of on-target/off-tumor toxicity seen by targeting TAAs that are also expressed in healthy cells such as EGFR or VEGFR [7].

Currently, an important number of bsAbs are being studied in many clinical trials, showing positive results in a specific group of tumoral cells and a prolonged antitumoral response. Particularly, some malignancies such as lymphomas seem to have a better antitumoral response with bsAbs, in comparison with myeloid neoplasias or solid tumors [8]. For solid tumors, an optimal antibody impregnation to the tumor has been reported; however, a short half-life and concerns about their safety are still subjects of study [9].

Despite the breakthrough that these bsAbs are represented in the field of cancer immunotherapy, there are still many questions to be answered and challenges to be solved about the safety, efficacy, and range of possible treatable tumors. Here, we review some basic aspects of the bsAbs and their functions, along with their common use in the current clinical practice.

## 2. Pharmacology of Bispecific Antibodies

Antibodies are molecules made of different structural and functional parts. These parts are combined to create molecules with unique affinity, specificity, and interaction properties. The special structure-function of these macromolecules is beyond the scope of this review; however, understanding some basic principles is essential to comprehend the pharmacology behind bsAbs [10].

From a structural point of view, a bsAbs is made of two identical light chains (LCs) and two identical heavy chains (HCs). Each domain, between the LC and HC has disulfide bonds. The structural conformation creates three zones: two antigen-binding fragments (Fab) and one crystallizable fragment or Fc. Both Fab regions bind to molecular targets, the same epitope. On the other hand, the Fc region attaches to receptors such as Fcγ receptors (FcγRs), C1q, and neonatal Fc receptor (FcRn), mediating its effector functions [10,11].

Previously, it was mentioned that although bsAbs contain two Fab regions, these only bind to the same epitope in an antigen, defining monospecificity. As the name suggests, bsAbs are bispecific, because they bind to two antigens. Structurally, there are two principal types of bsAbs: (1) single-chain variable fragment (scFv) antibodies and (2) full-length IgG-based antibody. In the past time, three techniques were used for their creation bispecific T-cell engager (BiTE), dual-affinity retargeting proteins (DARTs) and tandem diabodies (TandAbs) [12]. Currently, they are created by orthogonal Fab interface, DuoBody, XmAb, CrossMab, and knobs-into-holes (KiH) [3].

Mechanisms of action are diverse. First, the process of binding immune cells with tumor cells, leads to suppression of malignant cells’ ability to escape the immune response. Second, bsAbs decrease the expression of certain molecules and the release of immune suppression mediators. Additionally, bsAbs also block targets such as PD1, CTLA-4, LAG-3, IL-23, TNF-a, and others and they also stimulate immune cells, those mechanisms act synergistically [13,14].

From a pharmacokinetic (PK) view the properties of bsAbs oscillate due to their different compositions. The Fc region plays a key role in bsAbs PK. It is reported that bioavailability is lower by oral administration, so other ways of administration are better choices.

With respect of the distribution, three parameters affect this variable: extravasation, distribution within a tissue, and elimination/clearance. Extravasation occurs when macromolecules have a high volume of distribution (Vd) and tend to remain in the tissues. Additionally, it is known that intracellular catabolism and renal clearance, both influenced by Fc region, increase the molecular weight of some molecules leading to slow clearance. After that, those molecules bind to FcRn, escape acidic endosomes, and return to the circulation or to the interstitial space. In the case of specific elimination, bsAbs bind to specific antigens on cell surfaces mediating its clearance in a process named target-mediated drug disposition (TMDD). In an attempt to integrate some of the previous concepts, scFvs vs. full-length IgG-based bsAbs have different molecular weights; this distinction determines the route of clearance and the time in circulation and route of administration [15,16].

BsAbs pharmacology is complicated, many situations and variables affect its pharmacodynamics and pharmacokinetics. Understanding the design, development, and properties of these molecules is fundamental for the development of new molecules [14].

## 3. T-Cell Engaging Bispecific Antibodies

Among all the antibody-centered cancer therapy, T-cell engaging bispecific antibodies (T-biAbs) have a promising role in future cancer therapeutics. These antibodies use the main principle of two different binding arms of the bispecific antibodies. One of the binding sites recognizes the invariable CD3 subunits of cytotoxic T lymphocytes (CTLs) and the other one recognizes certain tumor antigens. Therefore, the T-biAb can activate CTLs bypassing the MHC pathway and redirecting this activation to attack specific cancer cells [17]. Currently, there are two FDA-approved T-biAbs blinatumomab (for the management of acute lymphoblastic leukemia) [18,19].

## 4. T-biAbs Action on Immune Effector Cells

In 1989 Gross and Eshhar et al. were the first ones to approach the concept of T-cell reactivation against tumor cells. At that time, they combined monoclonal antibodies and T-cell receptors to promote immune activation [20]. However, this early model had a lack of costimulatory signals, so T-cell redirection against tumor cells was not possible. Currently, a lot of options of T-bsAbs with costimulatory properties are available and are approved to treat cancer patients [5].

For a better understanding of the T-biAbs mechanism of action on immune effector cells, it is important to have in mind the usual adaptive immune response. T-cell activation depends on the linkage between the T-cell receptor (TCR)/CD3 complex and the major histocompatibility complex (MHC)/peptide complex and on the interaction of co-stimulatory receptors and their ligands [21]. In contrast, because of the strong affinity between T-BsAbs and TAA/CD3, a significant quantity of activation receptors (TCR/CD3 complexes) accumulate between cells, leading to an efficient T-cell activation with the need of only one receptor–ligand interaction [22,23]. However, some studies showed that those T cells activated by T-BsAbs are less effective over time because they experience more rapid exhaustion [24]. It is also reported that T-bsAbs activate memory T cells, including central memory and effector memory phenotypes, instead of naïve T cells that do not induce tumor lysis. Those memory T cells have highly cytotoxic activity on CD8-positive peripheral T cells because of the high gene expression associated with CD8+ T-cell function, resulting in a major anti-tumor activity [25].

Furthermore, some studies reported that T-bsAb increased T-cell proliferation in the tumor and recruitment from the periphery to tumor tissues, leading to a significant amount of tumor-infiltrating T cells after treatment [26,27,28]. It is known that CXCR3 receptor–ligand interaction regulates T-cell migration. Some studies have reported that T-bsAb treatment induces expression of pro-inflammatory cytokines and chemokines such as CXCR3 ligands in tumors and upregulation of CXCR3 expression on T cells, leading to a major T-cell migration [29].

It is important to highlight that our understanding of the exact mechanism of the activity of effector T cells engaged by T-bsAbs is still insufficient. For example, it is not fully understood if the presence of preexisting T cells inside the tumor is necessary for peripheral T-cells recruitment by T-bsAbs [30].

## 5. T-biAbs for Redirecting Cells of the Innate Immune System

As previously mentioned, bispecific antibodies are an intelligent strategy to put close IECs and their intended target cell. Antitumor effects can be provided directly or indirectly by innate or adaptive immune cells. Immune effector cells comprise an arsenal of supportive and cytotoxic cells that can arrest a threat from its beginning, clearing out infections or cells with early malignant properties, or at least slowing down the process while the adaptive immune system is appropriately recruited [31].

Effector mechanisms of the innate immune system can be enhanced by using biAbs [32]. Most data regarding biAbs in the immune system approach dendritic cells (DCs) or natural killer (NK) cells. DCs are professional antigen-presenting cells (APCs), with the highest efficacy in T-cell priming (initial activation of naïve T cells with its cognate MHC–epitope complex). For proper T-cell activation, TCR binding with MHC-Epitope complex is necessary; however, co-stimulatory signals also need to be present, namely the CD28/CD80-86 binding [33]. Primed DCs express high levels of CD80/86. BiAbs can be used to maximize the likelihood of encounter of DCs and T-cells when preserving the antibody Fc region.

In vitro experiments have shown that trifunctional biAbs were effective in inducing specific cytotoxicity of melanoma cells, with proper T-cell priming measured by the upregulation of CD69 and downregulation of CD62L (naïve T-cell marker). The rationale of trifunctional biAbs is to preserve the Fc region in the antibody structure [34]. When the Fc region is present, it can be used by innate immune cells to attach to the drug via FcγR and engage in the effector response. DCs will attach to the Fc region and get close to T-cells and their respective targets, inducing a more potent activation and cytotoxicity. However, these results have only been positive in vitro, where conditions can be easily controlled and isolated scenarios can be tested.

On the other hand, in the in vivo setting, biAbs with preserved Fc regions have shown to be counterproductive, as FcγR is also expressed by other innate immune system cells such as macrophages and NK cells, inducing overactivation of the immune system with further severe adverse events such as cytokine release syndrome (CRS) [35]. Moreover, by preserving Fc regions, the half-life of the antibody will be considerably decreased due to undesired attachment to other FcγR expressing cells. Silencing of Fc domains by mutating key peptides or using alternative scFv structures are current strategies used to bypass these effects [32,33,34,35].

NK cells have also been addressed in biAbs research. NKs are highly cytotoxic, less prone to exhaustion, and do not require binding to an MHC-Epitope complex. The standard effector mechanism exploited in monoclonal antibody therapy for cancer is antibody-dependent cellular cytotoxicity (ADCC). This exact mechanism has been shown to be critical in using biAbs, even in the absence of an Fc domain. NK cells are one of the most efficient IECs in surveying and executing tumor cell cytotoxicity. Engaging NK cells to malignant cells has been proven to be effective in reducing tumor size in animal models at a greater extent than regular mAb therapy [36]. Natural killer cell engagers (NKCEs), biAbs with a shortened and silenced Fc domain that engages NKp46 in NK cells to the desired target, are subjects of study. NKp46 is a vital membrane protein that triggers NK cytotoxicity in HLA class I-unprotected cells, a common phenomenon in cancer, where HLA class I is downregulated. The evidence suggests that treatment with these antibodies could increase the NK tumor infiltrating cells and generate a good tumor response in animal models [37].

Novel approaches with NK cells include engagers targeting CD16, especially the CD16A isoform (FcγRIIIA). Currently, most NK cell engagers are directed against antigens expressed in hematological malignancies (CD19, CD20, CD30, and CD33), a trend that is also seen in other immunotherapies such as CAR-T cells and CAR-NK cells [38,39,40,41]. An essential turn onto solid malignancies is necessary.

## 6. T-biAbs for the Restoration and Enhancement of Antitumor Immunity

Antitumor immunity is one the main mechanisms of cancer control under micro-homeostatic conditions. A wide array of cells, signaling peptides, and membrane-bound proteins are involved in the building machinery of surveillance for malignancy in healthy cells. CD8+ T cells and NK cells are the main orchestrators of this mechanism. Nevertheless, evolutionary events inside potentially malignant cells can induce transcriptomic regulatory mechanisms that might allow them to evade immune surveillance [42,43].

Some of these regulatory pathways include molecules such as CD80/86 and PD-L1, which respectively bind to CTLA-4 and PD1, two of the primary immune checkpoints. When attached to their receptors, these molecules can induce a switch off in T cells, making them dormant and further modifying the tumor microenvironment (TME), creating the perfect place for continued growth and progression. By restoring and enhancing these dormant cells, dramatic treatment responses can be achieved. These have been shown in the past decade with the introduction of multiple mAbs immune checkpoint inhibitors that target PD1, PD-L1, and CTLA-4 [42]. Important advancements in overall and progression-free survival have been achieved in lung cancer, renal cell carcinoma, urothelial carcinoma, and melanoma, among others [44].

The tumor microenvironment also induces important changes in other tumor-infiltrating cells, including macrophages, DCs, myeloid-derived suppressor cells, and plasma cells. From these cells, DCs are particularly important. It has been shown that reactivating DCs inside the tumor can restore CD8+ T-cell function and induce other anticancer changes in the TME [43].

Approaching immune restoration using biAbs is a relatively new field; most literature is recent and experimental, with promising results potentially translational into the clinic. Liu et al. performed an animal in vivo study using a biAb directed against CD3e (T-cells), and PD-L1. Analysis of their data showed that dramatic responses obtained in multiple syngeneic tumor models were due to restoring CD8+ T-cell activity when bsAbs were targeting PD-L1 in the membrane of DCs, inhibiting negative regulation and enhancing the stimulation via cytokines and co-stimulatory molecules. These data are intriguing as no tumor-specific antibody was used, but instead, a “generalist” approach of immune checkpoint blockade was done [45].

A similar approach was used by Kraman et al., using a biAb directed against LAG3 and PD-L1. LAG3 is another critical immune checkpoint and maker of T-cell exhaustion. Results showed high antitumoral activity in murine syngeneic models [46]. Cui et al. developed a biAb for PD-L1 and VEGF; this innovative approach showed high efficacy in preclinical models and exploited the association between high PD-L1 expression by tumor cells and the addiction to angiogenic pathways [47]. Ramaswamy et al. developed an anti-CD47/anti-PD-L1 biAb active against hematological malignancies [48]. Other researchers have tried not only to block PD-L1 but also to stimulate the effector cell at the same time by targeting a co-stimulatory molecule such as CD28 [49] or 4-1BB [50]; positive results were obtained in preclinical studies. All these new structures are made without a functional Fc domain, avoiding possible adverse events that did not allow previous biAbs to be approved for therapy in solid tumors.

## 7. Delivery Strategies for T-biAbs

Currently, T-biAbs are used as immunotherapy in cancer treatment. There are two approved delivery strategies for T-biAbs-based therapies: delivery of Y-biAbs as recombinant proteins or in vivo production.

### 7.1. Delivery of T-biAbs as Recombinant Proteins

T-biAbs as recombinant proteins are composed of two linked antibody-binding regions that simultaneously recognize two antigens [12]. One of the antigens recruits and activates T cells because it binds to the CD3ε chain of the T-cell receptor. The other antigen redirects the cytotoxicity of the T cells to tumor cells, because this antigen is on the surface of the tumor cells [51,52]. Additionally, a new strategy that has been developed eases tumor-specific antigen (TSA) and tumor-associated antigen (TAA) targeting by linking T-biAbs and chimeric antigen receptor T cells (CAR-T). These are known as switchable CAR-Ts and are activated by a bispecific protein [53,54].

In this delivery strategy, recombinant proteins with a short circulatory half-life (T-biAbs in tandem single-chain variable fragment recombinant format) and long circulatory half-life (T-biAbs with Fc domain format) are administered as continuous intravenous (IV) infusion or as repeated high- dose bolus injections respectively [52].

On this topic, it is worth highlighting that some limitations to this type of delivery have been reported. First of all, the high cost of therapy, is derived from the production process and the usually long treatment. Furthermore, it the difficulty to achieve sustained plasma levels, high enough to be effective and low enough to be not toxic [55]. Moreover, due to the different expression technologies to develop T-biAbs, the diverse glycosylation patterns might affect the efficacy of the T-biAbs [56]. Additionally, the poor stability during long-term storage and tendency to aggregate over time are also reported. Finally, some limitations related to systemic administration are noted. In some cases, an infusion pump is required for continuous delivery. In other cases, some purified antibodies need to be administered through slow IV infusions to limit infusion reactions [57].

### 7.2. Delivery of T-biAbs In Vivo Production

To seek the balance between efficacy and safety, in vivo gene therapy was developed. It is known that modified oncolytic viruses eliminate tumor cells and stimulate systemic immune response without harming healthy cells and tissues [52]. It is known that those viruses can be modified to encode therapeutic transgenes, such as a functional T-biAb leading to an in situ expression of T-biAb by tumor cells [58]. Currently, there are two ways to lead to in vivo secretion of T-biAbs. The first one is transduction by using vectors. The oncolytic viruses that have been engineered as vectors for T-biAb expression are adenovirus and measles [59]. The second one, is the in vivo inoculation of synthetic nucleic acid-encoded T-biAb by using messenger RNA and plasmid DNA [60]. Some studies have demonstrated that the in vivo delivery strategy has an effective antitumor activity (reducing tumor growth and metastasis) and leads to delayed cancer progression in mice models [56,57,61].

The mechanism of action of the latter delivery system has shown advantages over the former. One of the advantages is that it is a cheaper alternative because it obviates the need for manufacturing and administering purified T-biAbs. Additionally, in some cases the in vivo production maintains an effective antibody concentration, so no concerns about long-term storage and rapid renal clearance arise [51,61].

## 8. T-biAbs in Hematologic Malignancies

There is solid and growing evidence about the use of T-biAbs in hematologic malignancies. This is possible because most hematologic neoplasms fulfill two important features for an effective biAbs therapy. The first one is that antigens from hematologic malignancies are mainly (or only) expressed on malignant cells, leading to a reduction of on-target/off cancer toxicity. The second one is that the antigen is strongly associated with the malignant phenotype, leading to a reduction of antigen loss variants [62].

According to the anterior principle, the approach of T-biAbs focuses on targeting CD3 on T cells (CD3ε fragment) and an antigen commonly expressed in tumor cells. For example, for B-cell malignancies, some T-BiAbs target CD19 and CD20. CD19 persists the entire course of the B-cell development. This overexpressed target allows the lysis of malignant cells and avoids the attack of the normal lymphocytes [63].

The most remarkable drug of this kind is blinatumomab. This antibody is made of two scFv combined in one protein chain [8]. The history of this novel drug began in 2014 when it was approved for Philadelphia chromosome (Ph)-negative relapsed or refractory (R/R) B-cell precursor acute lymphoblastic leukemia (ALL) in adults. Later, in 2016, FDA approved blinatumomab for pediatric patients with Ph- R/R B-cell precursor ALL [8]. After that, the TOWER trial and the ALCANTARA trial gave support for the full approval of the drug for R/R B-cell precursor ALL in adults and children in 2017, and for minimal residual disease-positive B-cell precursor ALL in 2018 [64,65,66,67].

Besides the patients’ positive outcomes under blinatumomab treatment, some concerns related to the main adverse effects (AE) are still present. Among the most dangerous AE, neurological affections are the most common cause of interruption of the drug. Neurotoxicity symptoms include headache, tremor, confusion, disorientation, and other more life-threatening symptoms such as seizures or stupor. Other significant AEs include CRS, produced because of T cell and macrophages activation, and infectious and hematologic toxicity [68,69]. In addition to AEs, blinatumomab has a short half-life and a significant but limited response among R/R ALL patients, and this has exposed the need for newer therapeutic alternatives.

However, many other hematologic neoplasms could benefit from this kind of immunotherapy. Once again, by targeting a common antigen on the tumor cell’s surface, the therapeutic effort focuses on malignant cells with little or no on-target off-tumor toxicity. For example, for acute myeloid leukemia (AML), CD123 is one of the over-expressed antigens targeted by novel drugs such as flotetuzumab (MGD006) or vibecotamab (XmAb1405) with positive results in phase I/II and phase I studies respectively [70,71]. There is also growing evidence that molecules that block CD33, an acid-binding sialo-adhesins receptor expressed in nearly 90% of AML, such as gemtuzumab ozogamicin are safe and effective [72]. Among the molecules AMG330 (NCT02520427), AMG673 (NCT03224819), and AMV564 (NCT03144245) are being evaluated under clinical trials. Only to mention other possible targets currently under study for AML include FLT3, CLEC12A, and WT1 [73].

For other neoplasms as multiple myeloma (MM), some possible targets for immunotherapy include B-cell maturation antigen (BCMA), G protein-coupled resector 5D (GPR5D), CD38, and Fc receptor-like 5 (FCRL5) [73]. Anti-BCMA AMG 420 has demonstrated a response rate near 70% in patients with R/R MM in phase I trials with a favorable security profile [74]. Some molecules have been proposed for the other targets and are currently under evaluation [75,76]. For non-Hodgkin lymphoma ideal targets include CD19, CD20, and CD47, and for Hodking lymphoma CD30 targets seems promising, and for myelodysplasias syndromes CD123 could be a suitable targe but is still under discussion [8,73,77].

## 9. T-biAbs in Solid Malignancies

As it was mentioned earlier, since the approval of blinatumomab in 2014, T-biAbs have become an essential section in the investigation of hematologic malignancies. However, it is not the case for solid malignancies. There are some challenges in developing T-biAbs for solid tumors, one of them the hypoxia-induced immunosuppression in the tumor microenvironment, as a successful T-biAb depends on activation of CTL targeting CD3 [52].

Nevertheless, a large number of molecules that target antigens expressed on solid tumor malignant cells are under investigation (see Table 1). Targets under examination include epithelial cell adhesion molecule (EpCAM) in non-small cell lung carcinoma, glycan 3 (GPC3) in live cancer, HER2 in breast cancer, and prostate-specific membrane antigen (PSMA) in prostate cancer [9].

EpCAM is a conserved type I transmembrane protein, and it is highly expressed in some malignancies, lung cancer included [78]. Molecules such as catumaxomab demonstrated in phase I/II clinical trials a good response in treating malignant-related ascites. However, its production was stopped due to severe AEs (CRS and hepatotoxicity) [79,80]. Currently, MT110 (AMG110) demonstrated safety and tolerability, however, its clinical effect is still to be dilucidated [81].

In regards to GPC3, a heparan-sulfate proteoglycan expressed over the 65% of hepatocellular carcinoma and related to poor prognosis [82], is an important target as well. ERY74, a humanized IgG antibody, has demonstrated suppression of tumor growth that expresses GPC3 [83]. For HER2 in advanced breast cancer, the evidence is more solid. Options include ertumaxomab (HER2/CD3), amivantamab (JNJ-61186372), and MM-111 (HER2/HER3) have been proved to be safe and effective in human studies [9]. PSMA, expressed in androgen-independent prostate cancers, has several targeting antibodies. HPN424 (NCT03577028) and AMG160 (NCT3792841) also have shown positive responses in animal models and clinical scenarios but the complete understanding of the molecules is still to be shown in the future [84,85].

## 10. CAR-T Switches

Among other redirecting T-cell activation therapies against cancer cells, chimeric antigen receptors (CAR) expressed by T cells (CAR-T) represent a revolution in immunotherapy. The approval of tisagenlecleucel (tisa-cel) and axicabtagene ciloleucel (axi-cel) in 2017 by FDA marked a turning point in cancer treatment [86]. These cells are genetically medicated to express chimeric antigen receptor that enables them to target tumor cells.

CAR-Ts put together two of the three necessary signals to T-cell priming, the TCR/CD3 signals, and the costimulatory receptors/mediators. This allows the cell to recognize user-defined cell surface tumor-associated antigens. However, similar to other immune therapies, CAR-T cells have some AEs. Among them we can mention the off-target toxicity (attack of normal cells that express some common antigens as cancer cells), tumor lysis syndrome, and CRS.

A novel strategy called CAR-T switches has been developed to avoid these undesirable side effects that could be quite harmful. These switches focus on recruiting and activation of immune cells by a bispecific adapter protein. The adaptor, conformed as a monoclonal antibody, can target the tumor antigen and activate the dormant CAR-T with a protein located in one of its N- or C-terminal region [87].

These switches allow control of the CAR-T cells, improve their specificity, and avoid unpredictable reactivity. There are three perspectives to switch design: suicide, endogenous, and exogenous switches. The suicide switches are molecules exposed to CAR-T cells; they are activated by administering an external drug or molecular mediator (iC9, EGFR, HSV-TK, CD20). The endogenous switches comprehend molecular strategies of the immune synapsis such as target-antigen recognition or expression of immune regulation singles such as PD-1 or CTL-4 (i.e SynNotch switches). The exogenous switches include antibodies or derived proteins with multiple binding sites that could lead to a more specific union of the CRA-T cell with its target [88].

## 11. Conclusions

BiTAbs have been demonstrated to be a groundbreaking alternative in immune therapy and a vanguard option that combines the engineering of molecular biology and the biological activity of the immune system. These drugs represent a more individual and specific treatment for cancer patients providing a “personalized” attack on malignant cells. However, there is still a lot of information to be elucidated in relation to this kind of immunotherapy. The complexity of the pharmacokinetics and pharmacodynamics of these drugs represents a change for future application. Finally, the possible adverse effects related to the nature of its mechanism of action should be addressed in incoming studies. Nevertheless, the evidence is promising and growing, and maybe these pharmacological agents represent the next and safer generation for cancer treatment.

## Figures and Tables

**Figure 1 pharmaceutics-14-01243-f001:**
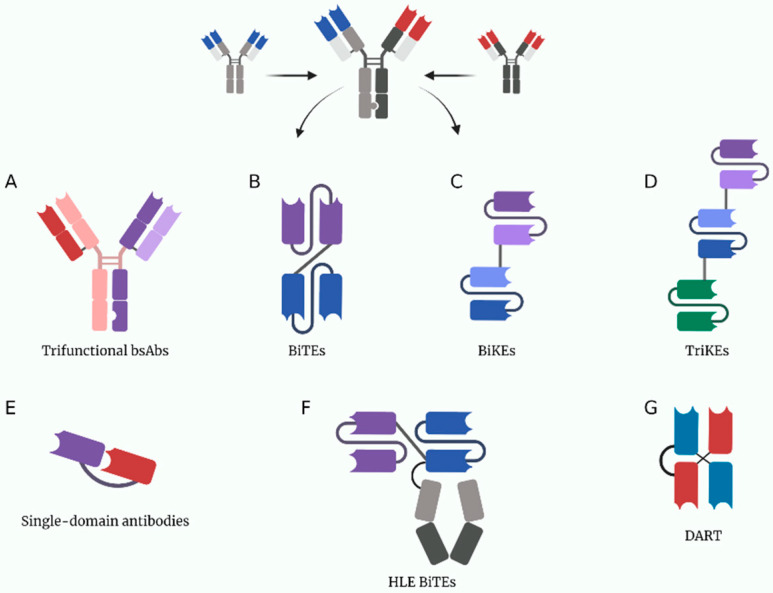
A depiction of some current multivalent antibody structures under study. (**A**) Trifuctional antibodies conserved their Fc domain to be able to bind to cells expressing Fc receptors. (**B**) BiTEs (bispecific T-cell engagers). (**C**) BiKEs (bispecific NK-cell engagers). (**D**) TriKEs (trispecific NK-cell engagers). (**E**) Single-domain antibodies only have one variable chain per target, they are usually made from heavy chain nanobodies derived from the structure of heavy-chain only camelid antibodies. (**F**) HLE BiTEs (half-life extended bispecific T-cell engagers) are BiTEs with an Fc portion that increases its half-life. (**G**) DARTs (dual affinity retargeting antibodies). Created with BioRender.com.

**Figure 2 pharmaceutics-14-01243-f002:**
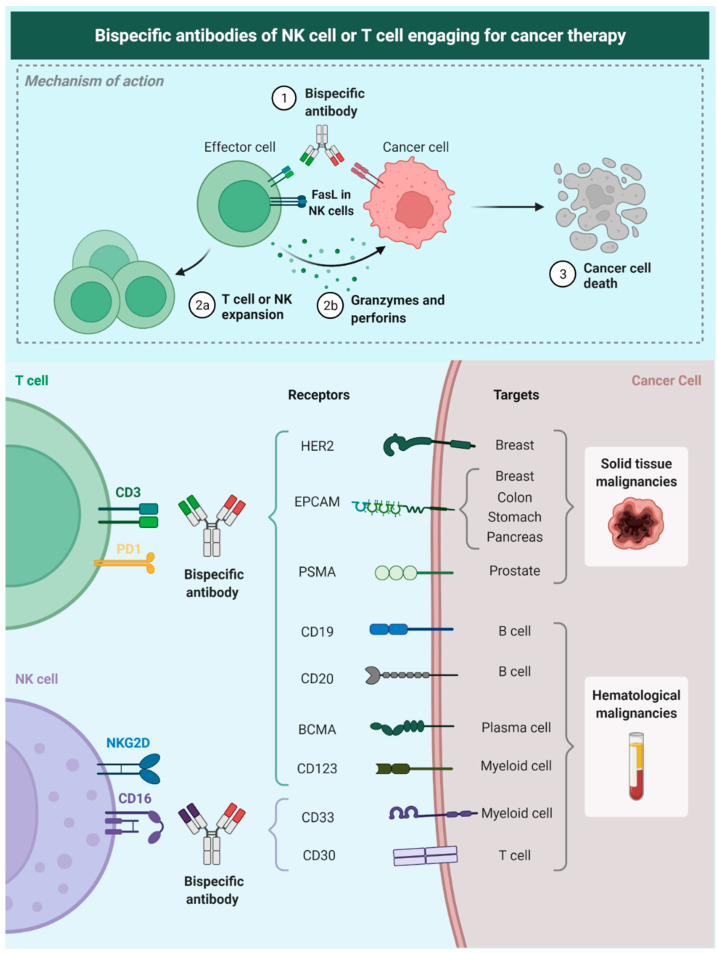
Description of the mechanism of NK-Bias; these antibodies target a tumor-related antigen and bind membrane receptors on NK cells allowing a spatial and molecular immune-mediated cell-killing process. We also show some of the tumor-associated targets that currently have been studied for therapy. Epithelial cell adhesion molecule (EpCAM), epidermal growth factor receptor 2 (HER2), prostate specific membrane antigen (PSMA), B-cell maturation antigen (BCMA), CD19, CD20, CD123, CD33, CD30 (cluster of differentiation [CD]).

**Table 1 pharmaceutics-14-01243-t001:** Some of the clinical trials of different molecules that remain under evaluation for use, we present the code of the trial (ClinicalTrials.gov accessed on 16 March 2022), the name of the trial, the type of cancer-related to the molecule, and the current phase of the study.

ClinicalTrials.gov Identifier	Title	Conditions	Interventions	Phase
**Recruiting**				
**NCT03146637**	Study of Activated CIK Armed With Bispecific Antibody for Advanced Liver Cancer	Advanced Liver Cancer	Biological: Activated CIK|Biological: CIK	Phase 2
**NCT05125016**	REGN4336 (a PSMAXCD3 Bispecific Antibody) Administered Alone or in Combination With Cemiplimab in Adult Male Patients With Metastatic Castration-Resistant Prostate Cancer	Metastatic Castration-resistant Prostate Cancer	Drug: REGN4336|Drug: Cemiplimab|Other: 18F-DCFPyL	Phase 2
**NCT04868877**	A Phase 1/2 Study Evaluating MCLA-129, a Human Anti-EGFR and Anti-c-MET Bispecific Antibody, in Patients With Advanced NSCLC and Other Solid Tumors	Non-Small Cell Lung Cancer Metastatic|Gastric Cancer|Head and Neck Cancer	Drug: MCLA-129	Phase 2
**NCT05090566**	MagnetisMM-4: Umbrella Study of Elranatamab (PF-06863135) in Combination With Anti-Cancer Treatments in Multiple Myeloma	Multiple Myeloma	Drug: Elranatamab + Nirogacestat|Drug: Elranatamab + lenalidomide + dexamethasone	Phase 2
**NCT03860207**	Study of the Safety and Efficacy of Humanized 3F8 Bispecific Antibody (Hu3F8-BsAb) in Patients With Relapsed/Refractory Neuroblastoma, Osteosarcoma and Other Solid Tumor Cancers	Neuroblastoma|Osteosarcoma|Other Solid Tumor Cancers	Biological: Humanized 3F8 Bispecific Antibody|Other: Blood draw	Phase 2
**NCT04380805**	A Study of AK104, a PD-1/CTLA-4 Bispecific Antibody in Subjects With Recurrent/Metastatic Cervical Cancer	Recurrent Cervical Cancer|Metastatic Cervical Cancer	Biological: AK104	Phase 2
**NCT04886271**	Recombinant Humanized Anti-CD47/PD-1 Bifunctional Antibody HX009 Injection in the Treatment of Advanced Solid Tumors	Advanced Solid Tumor	Drug: HX009	Phase 2
**NCT04276493**	Anti-HER2 Bispecific Antibody ZW25 Activity in Combination With Chemotherapy With/Without Tislelizumab	Breast Cancer|Gastric Cancer|Gastroesophageal Junction Cancer	Biological: ZW25|Drug: Docetaxel|Biological: Tislelizumab|Drug: Capecitabine|Drug: Oxaliplatin	Phase 2
**NCT04999605**	A Study of AK112 Combined With PARP Inhibitor in the Treatment of Recurrent Ovarian Cancer	Ovarian Neoplasms|Recurrent Ovarian Carcinoma|Relapsed Ovarian Cancer|Ovarian Cancer	Drug: AK112 low dose|Drug: AK112 medium dose|Drug: AK112 high dose	Phase 2
**NCT04618393**	A Study of EMB-02 in Participants With Advanced Solid Tumors	Advanced Solid Tumors	Biological: EMB-02	Phase 2
**NCT05214482**	A Study of AK112 in Advanced Malignant Tumors	Advanced Malignant Tumors	Drug: AK112|Drug: AK117|Drug: Chemotherapy	Phase 2
**NCT03406858**	Pembrolizumab and HER2Bi-Armed Activated T Cells in Treating Patients With Metastatic Castration Resistant Prostate Cancer	Castration Levels of Testosterone|Castration-Resistant Prostate Carcinoma|Prostate Carcinoma Metastatic in the Bone|PSA Progression|Stage IV Prostate Adenocarcinoma AJCC v7	Biological: HER2Bi-Armed Activated T Cells|Other: Laboratory Biomarker Analysis|Biological: Pembrolizumab	Phase 2
**NCT02912949**	A Study of Zenocutuzumab (MCLA-128) in Patients With Solid Tumors Harboring an NRG1 Fusion	Solid Tumours Harboring NRG1 Fusion|NSCLC Harboring NRG1 Fusion|Pancreatic Cancer Harboring NRG1 Fusion|NRG1 Fusion	Drug: zenocutuzumab (MCLA-128)	Phase 2
**NCT04995523**	A Study to Assess the Safety and Efficacy of AZD2936 in Participants With Advanced or Metastatic Non-small Cell Lung Cancer (NSCLC)	Non-Small-Cell Lung Carcinoma	Drug: AZD2936	Phase 2
**NCT05102214**	HLX301 (TIGIT × PDL1 Bispecific) in Patients With Locally Advanced or Metastatic Solid Tumors	Locally Advanced or Metastatic Solid Tumors|Non-small Cell Lung Cancer	Drug: HLX301	Phase 2
**NCT03269526**	BATs Treatment for Pancreatic Cancer, Phase Ib/II	Locally Advanced Pancreatic Adenocarcinoma|Metastatic Pancreatic Adenocarcinoma	Drug: EGFR BATs after standard of care chemo	Phase 2
**NCT04547101**	A Study of AK104 in Subjects With Locally Advanced Unresectable or Metastatic MSI-H/dMMR Solid Tumors	MSI-H/dMMR Solid Tumor	Drug: AK104	Phase 2
**NCT03564340**	Study of REGN4018 Administered Alone or in Combination With Cemiplimab in Adult Patients With Recurrent Ovarian Cancer	Recurrent Ovarian Cancer|Recurrent Fallopian Tube Cancer|Recurrent Primary Peritoneal Cancer	Drug: REGN4018|Drug: cemiplimab	Phase 2
**NCT05159388**	A Study of PRS-344/S095012 (PD-L1 × 4-1BB Bispecific Antibody-Anticalin Fusion) in Patients With Solid Tumors	Solid Tumor	Drug: PRS-344/S095012	Phase 2
**NCT04626635**	REGN7075 in Combination With Cemiplimab in Adult Participants With Advanced Solid Tumors	Advanced Solid Tumors	Drug: REGN7075|Drug: cemiplimab	Phase 2
**NCT04930432**	Study of MCLA-129, a Human Bispecific EGFR and cMet Antibody, in Patients With Advanced NSCLC and Other Solid Tumors	Solid Tumor|Non-Small Cell Lung Cancer|Head and Neck Cancer|Colorectal Cancer	Drug: MCLA-129	Phase 2
**NCT04750239**	Safety and Clinical Activity of Nivatrotamab in Relapsed/Recurrent Metastatic Small-cell Lung Cancer	SCLC	Drug: Nivatrotamab	Phase 2
**NCT04931654**	A Study to Assess the Safety and Efficacy of AZD7789 in Participants With Advanced or Metastatic Solid Cancer	Carcinoma, Non-Small-Cell Lung	Drug: AZD7789	Phase 2
**NCT03440437**	FS118 First in Human Study in Patients With Advanced Malignancies	Advanced Cancer|Metastatic Cancer|Squamous Cell Carcinoma of Head and Neck	Drug: FS118	Phase 2
**NCT04900363**	A Trial of AK112 (PD-1/VEGF Bispecific Antibody) in Patients With NSCLC	Non-small Cell Lung Cancer	Drug: AK112	Phase 2
**NCT04870177**	Study of AK112 in the Treatment of Advanced Gynecological Tumors	Gynecologic Cancer|Cancer Metastatic|Ovarian Neoplasms|Cervical Neoplasm|Endometrial Neoplasms	Drug: AK112	Phase 2
**NCT04557098**	A Study of Teclistamab, in Participants With Relapsed or Refractory Multiple Myeloma	Hematological Malignancies	Drug: Teclistamab	Phase 2
**NCT04634552**	A Study of Talquetamab in Participants With Relapsed or Refractory Multiple Myeloma	Hematological Malignancies	Drug: Talquetamab	Phase 2
**NCT05180474**	Research Trial to Study Safety of GEN1047 (DuoBody^®^-CD3xB7H4) in Participants With Malignant Solid Tumors	Breast Cancer|Uterine Cancer|Ovarian Cancer|Non Small Cell Lung Cancer (NSCLC)|Cervical Cancer|Head and Neck Squamous Cell Carcinoma (HNSCC), Except for Nasopharyngeal Carcinoma|Urothelial Cancer|Cholangiocarcinoma (CCA)	Biological: GEN1047 is a bispecific antibody that induces T-cell-mediated cytotoxicity of B7H4-positive cells.	Phase 2
**NCT03888105**	Assess the Anti-Tumor Activity and Safety of Odronextamab in Patients With Relapsed or Refractory B-cell Non-Hodgkin Lymphoma	B-cell Non-Hodgkin Lymphoma (NHL)	Drug: Odronextamab	Phase 2
**NCT04696809**	A Study of Teclistamab in Japanese Participants With Relapsed or Refractory Multiple Myeloma	Hematologic Malignancies	Drug: Teclistamab	Phase 2
**NCT04496674**	Bispecific PSMAxCD3 Antibody CC-1 in Patients With Squamous Cell Carcinoma of the Lung	Lung Cancer Squamous Cell	Drug: CC-1 and Toczilizumab	Phase 2
**NCT05228470**	MagnetisMM-8: Study Of Elranatamab (PF-06863135) Monotherapy in Chinese Participants With Refractory Multiple Myeloma	Elranatamab|Myeloma|Multiple Myeloma|Relapsed Multiple Myeloma|Refractory Multiple Myeloma|PF-06863135|BCMA|Bispecific|Bispecific Antibody|BCMA-CD3 Bispecific|MagnetisMM-8	Drug: Elranatamab	Phase 2
**NCT04590781**	Safety and Efficacy of XmAb18087 ± Pembrolizumab in Advanced Merkel Cell Carcinoma or Extensive-stage Small Cell Lung Cancer	Merkel Cell Carcinoma|Small Cell Lung Cancer	Biological: XmAb18087|Drug: XmAb18087 ± Pembrolizumab	Phase 2
**NCT03272334**	Her2-BATS and Pembrolizumab in Metastatic Breast Cancer	Metastatic Breast Cancer	Drug: HER2 BATs with Pembrolizumab	Phase 2
**NCT04492033**	A Study of ABL001 in Combination With Irinotecan or Paclitaxel in Patients With Advanced or Metastatic Solid Tumors	P1b: Advanced Solid Tumors|P2: Biliary Tract Cancer	Drug: ABL001|Drug: Paclitaxel|Drug: Irinotecan	Phase 2
**NCT04466891**	A Study of ZW25 (Zanidatamab) in Subjects With Advanced or Metastatic HER2-Amplified Biliary Tract Cancers	HER2-amplified Biliary Tract Cancers	Drug: ZW25 (Zanidatamab)|Diagnostic Test: In situ hybridization (ISH)-based companion diagnostic assay|Diagnostic Test: Immunohistochemistry (IHC)-based companion diagnostic assay	Phase 2
**NCT03929666**	A Safety and Efficacy Study of ZW25 (Zanidatamab) Plus Combination Chemotherapy in HER2-expressing Gastrointestinal Cancers, Including Gastroesophageal Adenocarcinoma, Biliary Tract Cancer, and Colorectal Cancer	HER2-expressing Gastrointestinal Cancers, Including Gastroesophageal Adenocarcinoma, Biliary Tract Cancer, and Colorectal Cancer	Drug: ZW25 (Zanidatamab)|Drug: Capecitabine|Drug: Cisplatin|Drug: Fluorouracil|Drug: Leucovorin|Drug: Oxaliplatin|Drug: Bevacizumab|Drug: Gemcitabine	Phase 2
**NCT03761108**	First in Human (FIH) Study of REGN5458 in Patients With Relapsed or Refractory Multiple Myeloma	Multiple Myeloma	Drug: REGN5458	Phase 2
**NCT04224272**	A Study of ZW25 (Zanidatamab) With Palbociclib Plus Fulvestrant in Patients With HER2+/HR+ Advanced Breast Cancer	HER2+/HR+ Breast Cancer	Drug: ZW25 (Zanidatamab)|Drug: Palbociclib|Drug: Fulvestrant	Phase 2
**NCT05176665**	EMB-01 in Patients With Advanced/Metastatic Gastrointestinal Cancers	Neoplasms|Neoplasm Metastasis|Metastatic Gastrointestinal Carcinoid Tumor	Drug: EMB-01	Phase 2
**NCT03797391**	A Dose Escalation With Expansion Study of EMB-01 in Participants With Advanced/Metastatic Solid Tumors	Neoplasms|Neoplasm Metastasis|Non-Small-Cell Lung Cancer	Drug: EMB-01	Phase 2
**NCT04785820**	A Study of RO7121661 and RO7247669 Compared With Nivolumab in Participants With Advanced or Metastatic Squamous Cell Carcinoma of the Esophagus	Advanced or Metastatic Esophageal Squamous Cell Carcinoma	Drug: RO7121661|Drug: RO7247669|Drug: Nivolumab	Phase 2
**NCT04735575**	A Ph1/2 Study of EMB-06 in Participants With Recurrent or Refractory Myeloma	Relapsed or Refractory Multiple Myeloma	Biological: EMB-06	Phase 2
**NCT05014412**	A Study to Learn About the Study Medicine (Elranatamab) in Participants With Multiple Myeloma That Has Come Back After Responding to Treatment or Has Not Responded to Treatment	Multiple Myeloma	Drug: Elranatamab	Phase 2
**NCT05189093**	Recombinant Humanized Anti-CD47/PD-1 Bifunctional Antibody HX009 in Patients With Relapsed/Refractory Lymphoma	Relapsed/Refractory Lymphoma	Drug: Recombinant humanized anti-CD47/PD-1 bifunctional antibody HX009 injection	Phase 2
**NCT04728321**	A Study of Anti-PD-1/CTLA-4 Bispecific AK104 Alone or in Combination With Lenvatinib in Advanced Hepatocellular Carcinoma	Hepatocellular Carcinoma	Biological: AK104 lenvatinib|Biological: AK104	Phase 2
**NCT04889716**	CAR-T Followed by Bispecific Antibodies	Large B-cell Lymphoma	Drug: mosunetuzumab|Drug: glofitamab|Drug: obinutuzumab	Phase 2
**NCT04444167**	A Study of Anti-PD-1/CTLA-4 Bispecific AK104 Plus Lenvatinib in First-line Advanced Hepatocellular Carcinoma	Hepatocellular Carcinoma	Biological: AK104|Drug: Lenvatinib	Phase 2
**NCT04444141**	A Study of PD-1/CTLA-4 Bispecific AK104 in Relapsed or Refractory Peripheral T-cell Lymphoma	Peripheral T-cell Lymphoma	Biological: AK104	Phase 2
**NCT04602065**	Evaluation of Safety and Efficacy of IBI318 Monotherapy for Relapsed/Refractory Extranodal NK/T Cell Lymphoma (Nasal Type) Trial	Extranodal NK/T Cell Lymphoma, Nasal Type	Drug: IBI318(Recombinant human anti-PD1/PD-L1 bispecific antibody)	Phase 2
**NCT05044897**	A Clinical Study to Evaluate the Efficacy and Safety of SI-B001 in the Treatment of Recurrent and Metastatic HNSCC	Head and Neck Squamous Cell Carcinoma	Drug: SI-B001	Phase 2
**NCT04763083**	First in Human Study of NVG-111 in Chronic Lymphocytic Leukaemia and Mantle Cell Lymphoma	Chronic Lymphocytic Leukaemia|Small Lymphocytic Lymphoma|Mantle Cell Lymphoma	Drug: NVG-111|Drug: NVG-111 (RP2D)	Phase 2
**NCT04703686**	Treatment by a Bispecific CD3xCD20 Antibody for Relapse/Refractory Lymphomas After CAR T-cells Therapy	Diffuse Large B-Cell Lymphoma Refractory|Refractory Indolent Adult Non-Hodgkin Lymphoma|Refractory Transformed B-cell Non-Hodgkin Lymphoma|Refractory Primary Mediastinal Large B-Cell Cell Lymphoma|Refractory Mantle Cell Lymphoma	Drug: Obinutuzumab|Drug: RO7082859	Phase 2
**NCT04469725**	KN046 (a Humanized PD-L1/CTLA4 Bispecific Single Domain Fc Fusion Protein Antibody) in Subjects With Thymic Carcinoma	Thymic Carcinoma	Drug: KN046	Phase 2
**NCT05020236**	MagnetisMM-5: Study of Elranatamab (PF-06863135) Monotherapy and Elranatamab + Daratumumab Versus Daratumumab + Pomalidomide + Dexamethasone in Participants With Relapsed/Refractory Multiple Myeloma	Multiple Myeloma	Drug: Elranatamab|Drug: Daratumumab|Drug: Pomalidomide|Drug: Dexamethasone	Phase 3
**NCT05152147**	A Study of Zanidatamab in Combination With Chemotherapy Plus or Minus Tislelizumab in Patients With HER2-positive Advanced or Metastatic Gastric and Esophageal Cancers	Gastric Neoplasms|Gastroesophageal Adenocarcinoma|Esophageal Adenocarcinoma	Drug: Zanidatamab|Drug: Tislelizumab|Drug: Trastuzumab|Drug: Capecitabine|Drug: Oxaliplatin|Drug: Cisplatin|Drug: 5-Fluorouracil|Diagnostic Test: In situ hybridization (ISH)-based companion diagnostic assay|Diagnostic Test: Immunohistochemistry (IHC)-based companion diagnostic assay	Phase 3
**NCT03643276**	Treatment Protocol for Children and Adolescents With Acute Lymphoblastic Leukemia—AIEOP-BFM ALL 2017	Acute Lymphoblastic Leukemia, Pediatric	Drug: Blinatumomab|Drug: Bortezomib|Drug: Cyclophosphamide|Drug: Cytarabine|Drug: Daunorubicin|Drug: Myocet|Drug: Dexamethasone|Drug: Doxorubicin|Drug: Etoposide|Drug: Fludarabine Phosphate|Drug: Ifosfamide|Drug: 6-Mercaptopurine|Drug: Methotrexate|Drug: Pegaspargase|Drug: Prednisolone|Drug: Tioguanin|Drug: Vincristine|Drug: Vindesine|Drug: Erwinase	Phase 3
**NCT04722848**	Sequential Treatment With Ponatinib and Blinatumomab vs. Chemotherapy and Imatinib in Newly Diagnosed Adult Ph+ ALL	Acute Lymphoblastic Leukemia (Philadelphia Chromosome Positive)|ALL, Adult|Philadelphia-Positive ALL	Drug: Ponatinib + Blinatumomab|Drug: Chemotherapy + Imatinib	Phase 3
**Active, not recruiting**				
**NCT02620865**	Bispecific Antibody Armed Activated T-cells With Aldesleukin and Sargramostim in Treating Patients With Locally Advanced or Metastatic Pancreatic Cancer	Metastatic Pancreatic Adenocarcinoma|Recurrent Pancreatic Carcinoma|Stage III Pancreatic Cancer|Stage IV Pancreatic Cancer	Biological: Aldesleukin|Biological: Antibody Therapy|Drug: Fluorouracil|Drug: Gemcitabine Hydrochloride|Drug: Irinotecan Hydrochloride|Other: Laboratory Biomarker Analysis|Drug: Leucovorin Calcium|Drug: Oxaliplatin|Drug: Paclitaxel Albumin-Stabilized Nanoparticle Formulation|Biological: Sargramostim	Phase 2
**NCT03321981**	MCLA-128 With Trastuzumab/Chemotherapy in HER2+ and With Endocrine Therapy in ER+ and Low HER2 Breast Cancer	Breast Cancer Metastatic	Drug: MCLA-128|Drug: Trastuzumab|Drug: Vinorelbine|Drug: Endocrine therapy	Phase 2
**NCT04649359**	MagnetisMM-3: Study Of Elranatamab (PF-06863135) Monotherapy in Participants With Multiple Myeloma Who Are Refractory to at Least One PI, One IMiD and One Anti-CD38 mAb	Multiple Myeloma	Drug: Elranatamab (PF-06863135)	Phase 2
**NCT04083534**	First In Human (FIH) Study of REGN5459 in Adult Patients With Relapsed or Refractory Multiple Myeloma (MM)	Relapsed Multiple Myeloma|Refractory Multiple Myeloma	Drug: REGN5459	Phase 2
**Not yet recruiting**				
**NCT04868708**	A Study of AK104 (an Anti-PD-1 and Anti-CTLA-4 Bispecific Antibody) in Recurrent or Metastatic Cervical Cancer	Recurrent or Metastatic Cervical Cancer	Biological: AK104|Biological: Bevacizumab|Drug: Paclitaxel|Drug: Cisplatin or Carboplatin	Phase 2
**NCT04556253**	AK104 in Locally Advanced MSI-H/dMMR Gastric Carcinoma and Colorectal Cancer	MSI-H/dMMR Gastric Carcinoma and Colorectal Cancer	Drug: AK104	Phase 2
**NCT04172454**	Safety and Efficacy of AK104, a PD-1/CTLA-4 Bispecific Antibody, in Selected Advanced Solid Tumors	Advanced Solid Tumors|Melanoma	Biological: AK104	Phase 2
**NCT04597541**	A Study of AK112, a PD-1/VEGF Bispecific Antibody, for Advanced Solid Tumors	Solid Tumor, Adult	Drug: AK112	Phase 2
**NCT05229497**	A Phase Ib/II Study of AK112 in Combination With AK117 in Advanced Malignant Tumors	Advanced Malignant Tumors	Drug: AK112|Drug: AK117|Drug: Carboplatin|Drug: Cisplatin|Drug: 5-Fluorouracil	Phase 2
**NCT05235542**	A Phase Ib/II Study of AK104 and AK117 in Combination With or Without Chemotherapy in Advanced Malignant Tumors	Advanced Malignant Tumors	Drug: AK104|Drug: AK117|Drug: Capecitabine tablets|Drug: Oxaliplatin|Drug: Cisplatin|Drug: Paclitaxel|Drug: Irinotecan|Drug: Docetaxel|Drug: 5-FU	Phase 2
**NCT05247684**	AK112 Neoadjuvant/Adjuvant Treatment for Resectable NSCLC	Resectable Non-small Cell Lung Cancer	Drug: AK112|Drug: Carboplatin|Drug: Cisplatin|Drug: Paclitaxel	Phase 2
**NCT04841538**	A Study of ES101 (PD-L1x4-1BB Bispecific Antibody) in Patients With Advanced Malignant Thoracic Tumors	Thoracic Tumors|Non-small Cell Lung Cancer|Small Cell Lung Cancer	Drug: ES101	Phase 2
**NCT05227651**	AK104 in Neoadjuvant Treatment of Cervical Cancer	Cervical Cancer	Drug: AK104	Phase 2
**NCT05215067**	A Phase II Trial of AK104 in Advanced Non-Small Cell Lung Cancer	Advanced Non-small Cell Lung Cancer	Drug: AK104|Drug: Docetaxel	Phase 2
**NCT05032040**	A Study of XmAb20717 in Patients With Selected Advanced Gynecologic and Genitourinary Malignancies	Ovarian Cancer|Clear Cell Carcinoma|Endometrial Cancer|Cervical Carcinoma|Metastatic Castration-Resistant Prostate Cancer (mCRPC)	Biological: XmAb20717	Phase 2
**NCT05216835**	Safety and Preliminary Efficacy Assessment of AZD7789 in Patients With Relapsed or Refractory Classical Hodgkin Lymphoma	Relapsed or Refractory Classical Hodgkin Lymphoma	Drug: AZD7789	Phase 2
**NCT04542837**	The Study of KN046 in Combination With Lenvatinib in Advanced Hepatocellular Carcinoma	HCC	Biological: KN046|Drug: Lenvatinib	Phase 2
**NCT05256472**	A Study of AK104 Plus Axitinib in Advanced/Metastatic Clear Cell Renal Cell Carcinoma	Clear Cell Renal Cell Carcinoma|First-line Treatment	Drug: AK104|Drug: axitinib	Phase 2
**Unknown status**				
**NCT03852251**	A Study of AK104, a PD-1/CTLA-4 Bispecific Antibody, for Advanced Solid Tumors or With mXELOX as First-line Therapy for Advanced Gastric or GEJ Adenocarcinoma	Gastric Adenocarcinoma|Advanced Solid Tumors|Gastroesophageal Junction Adenocarcinoma	Biological: AK104|Drug: Oxaliplatin|Drug: Capecitabine	Phase 2
**NCT02173093**	Activated T Cells Armed With GD2 Bispecific Antibody in Children and Young Adults With Neuroblastoma and Osteosarcoma	Disseminated Neuroblastoma|Recurrent Neuroblastoma	Biological: IL-2|Biological: GD2Bi-aATC|Biological: GM-CSF|Other: laboratory evaluations of immune responses	Phase 2
**NCT04220307**	A Study of a PD-1/CTLA-4 Bispecific Antibody AK104 in Patients With Metastatic Nasopharyngeal Carcinoma	Nasopharyngeal Carcinoma	Biological: AK-104	Phase 2
**NCT02744768**	D-ALBA Frontline Sequential Dasatinib and Blinatumomab in Adult Philadelphia Positive Acute Lymphoblastic Leukemia	Acute Lymphoblastic Leukemia	Drug: Dasatinib|Drug: Blinatumomab	Phase 2

## Data Availability

Not applicable.

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
