# Peer review of "Bispecific Antibodies in Cancer Immunotherapy: A Novel Response to an Old Question"

_pharmaceutics, 2022, doi:10.3390/pharmaceutics14061243_

Round 1

Reviewer 1 Report

This is a fine review about bispecific antibodies. The authors only discuss the use of such agents to bridge recognition of TAAs with T cells or NK cells. They have missed another strategy, namely to target with one arm a viral antigen from an oncolytic virus and with the other arm T cells or NK cells. A review of this concept was published in BioDrugs 2013. Doi: 10.1007/s40259-012-0008-z.

The paper should be revised accordingly.

Author Response

Point 1. This is a fine review about bispecific antibodies. The authors only discuss the use of such agents to bridge recognition of TAAs with T cells or NK cells. They have missed another strategy, namely to target with one arm a viral antigen from an oncolytic virus and with the other arm T cells or NK cells. A review of this concept was published in BioDrugs 2013. Doi: 10.1007/s40259-012-0008-z.

The paper should be revised accordingly.

Response 1. Thank you for your kind suggestion. We reviewed this part appropriately in line 90, citing the document and explaining the approach of combining oncolytic virus therapy with bsAbs.

Reviewer 2 Report

See the attached PDF file

Author Response

Point 1. To Table 1: Add the identity of the antibody targets. Especially since on page 9 starting line 394, the authors write: “Nevertheless, an important number of molecules target some antigens expressed on solid tumor malignant cells (see Table 1)”. (But none of the targets is identified in Table 1). Moreover, I would recommend arranging Table 1.

Response 1. We thank the reviewer for the suggestion. We changed the phrase starting in line 394 to “Nevertheless, a large number of molecules that target antigens expressed on solid tumor malignant cells are under investigation”; in order to avoid any confusion or misunderstanding in the readers.

Point 2. English language editing required

Response 2. We thank the reviewer for the suggestion. We made an extensive review of the whole manuscript.

Point 3. In the list of References, the numbering of the references is duplicated.

Response 3. We thank the reviewer for the suggestion. The segment was modified according to the recommendation of the reviewer.

Point 4. Page 4 line 163: the sentence: “Currently, there are two approved T-biAbs: catumaxomab (used in the treatment of malignant ascites) . . .” is not up-to date. Catumaxomab was approved only in Europe in 2009 (was not approved by the FDA) and withdrawn in 2017.
On page 9 line 401 the authors write that “ . . . its production was stopped due to severe AEs (CRS and hepatotoxicity)[79]”. Reference 79 is from the phase II/III clinical trial of catumaxomab, published in 2010, which does not disclose the official withdrawal of catumaxomab from the market.

Response 4. We removed the text portion talking about catumaxomab. We narrowed it down to only blinatumomab, as it is currently the only T-cell engager bsAb approved. For the second query, we added an appropriate reference in which catumaxomab withdrawal was addressed.

Point 5. Starting on page 7 line 321, there is a section on in vivo production of bispecific antibodies. This section fails to discuss major limitations of the in vivo approaches such as no control over expression level and on terminating antibody production when no longer required. On page 8 starting line 336 there is a sentence: “Additionally, the in vivo production maintains an effective antibody concentration, so no concerns about long-term storage and rapid renal clearance arise[62].” Reference 62 is from 2004, not very up to date. With bsAbs being a relatively young and fast-evolving field, I would expect more recent citations. An additional “old” and out-dated” reference is 25. yo

Response 5. We thank the reviewer for the suggestions. We removed the reference 25 and updated the reference 62, now 61. About the other recommendatoin, we suggest to leave the section like we wrote it, because it would be a very extensive manuscript if we change it. However, if is necessary we could add this section.

Point 6. Figure 1 describes “ . . . some of the most common design strategies of BiAbs currently manufactured in the industry”. The review and Figure 1 falls short of discussing (even briefly) the multitude of bsAb formats (and how important format is for efficacy), that can be found in excellent recent reviews by leaders of the field such as Ronald Kontermann and Ulrich Brinkmann (for example: https://www.science.org/doi/10.1126/science.abg1209?url_ver=Z39.88- 2003&rfr_id=ori:rid:crossref.org&rfr_dat=cr_pub%20%200pubmed). And https://pubmed.ncbi.nlm.nih.gov/32542108/  .

Response 6. Figure was remade. We added other critical structures used currently for bispecific antibodies.

Point 7. page 2 line 83 change “liufe” to “life”

Response 7. We agree with the reviewer's suggestion. We have changed “luife” to“life” in page 2 line 83.

Point 8. page 3 line 114 change “Fcy” to “Fcg” (the symbol “gamma”. Same on page 5 line 228.

Response 8. We agree with the reviewer's suggestion. We have changed “Fcy” to “Fcγ” (the symbol “gamma”) in page 3 line 83, in page 5 line 228 and in the whole text.

Point 9. Check spelling and wording of References carefully: on page 31 line 644 change “Ef Fi Cacy” with “Efficacy”.

Response 9. We agree with the reviewer's suggestion. We have changed “Ef Fi Cacy” to“Efficacy” in page 31 line 644.

Reviewer 3 Report

This paper discussed some basic aspects of the design and function of bsAbs, their main challenges and the state of the art of these molecules in the treatment of hematological and solid malignancies, and future perspectives. Although this paper is overall well written, there are several grammatical mistakes and typos that have to be corrected.

Line 53, "...solid malignancies..." instead of "...solid malignances..."

Line 85, “…expressed on the surface of …” instead of “…expressed in the surface of …”

Line 93, “… seem to…” instead of “…seems to…”

Line 100, “…we can use…” instead of “…we can used…”

Line 118, “they attack two antigens” instead of “they attacks two antigens”

Line 126, “to escape…” instead of “to scape…”

Line 150, “pharmacodynamics” instead of “pharmacodynamic”

Line 151, “…these molecules…” instead of “…this molecules…”

Line 217, “…is to preserve…” instead of “…is preserving…”

Line 251, please correct the sentence as “Antitumor immunity is the main mechanism of …”

Line 278, please correct the sentence as ” …stopping the negative regulation and enhancing…”

Line 305, “These are known as…” instead of “These is known as…”

Line 315, “to develop” instead of “ to develope”

Line 329, please correct the sentence as ”And the second one is the direct…”

Line 342, please correct the sentence as ”…on malignant cells and that antigens is…”

Line 346, “…CD19 and CD20” instead of “ …CD19 an CD20”

Line 390, “…solid malignancies” instead of “…solid malignances”

Line 391, “…hypoxia-induced immunosuppression…” instead of “…hypoxic-induced immunosuppression…”

Line 408, “…expresses GPC3” instead of “…express GPC3”

Line 432, “improve their specificity” instead of “improve its specificity”

Line 439, “lead to a more …” instead of “lead a more…”

Line 442, please correct the sentence as “BiTAbs have been demonstrated…”

Author Response

This paper discussed some basic aspects of the design and function of bsAbs, their main challenges and the state of the art of these molecules in the treatment of hematological and solid malignancies, and future perspectives. Although this paper is overall well written, there are several grammatical mistakes and typos that have to be corrected.

Point 1. Line 53, "...solid malignancies..." instead of "...solid malignances..."

Response 1. We agree with the reviewer's suggestion. We have changed "...solid malignances..." in line 53.

Point 2. Line 85, “…expressed on the surface of …” instead of “…expressed in the surface of 

Response 2. We agree with the reviewer's suggestion. We have changed "... expressed in the surface of ." to  “…expressed on the surface of …” in line 85.

Point 3. Line 93, “… seem to…” instead of “…seems to…”

Response 3. We agree with the reviewer's suggestion. We have changed "... seems to” to “ …seem to..” in line 93

Point 4. Line 100, “…we can use…” instead of “…we can used…”

Response 4. We agree with the reviewer's suggestion. We have changed "... we can used..” to “ …we can use…” in line 100.

Point 5. Line 118, “they attack two antigens” instead of “they attacks two antigens”

Response 5. We agree with the reviewer's suggestion. We have changed “they attacks two antigens”  to “they attack two antigens” in line 118.

Point 6. Line 126, “to escape…” instead of “to scape…”

Response 6. We agree with the reviewer's suggestion. We have changed “to scape” to “to escape” in line 126.

Point 7. Line 150, “pharmacodynamics” instead of “pharmacodynamic”

Response 7. We agree with the reviewer's suggestion. We have changed “pharmacodynamic” to “pharmacodynamics” in line 150.

Point 8. Line 151, “…these molecules…” instead of “…this molecules…”

Response 8. We agree with the reviewer's suggestion. We have changed “…this molecules…” to “…these molecules…” in line 151.

Point 9. Line 217, “…is to preserve…” instead of “…is preserving…”

Response 9. We agree with the reviewer's suggestion. We have changed “…is preserving…” to  “…is to preserve…” in line 217.

Point 10. Line 251, please correct the sentence as “Antitumor immunity is the main mechanism of …”

Response 10. We agree with the reviewer's suggestion. We have changed “Antitumor immunity is the main mechanism” to “Antitumor immunity is ONE of the main”

Point 11. Line 278, please correct the sentence as ” …stopping the negative regulation and enhancing…”

Response 11. We agree with the reviewer's suggestion. We have changed “stopping negative regulation” to “inhibiting negative regulation and enhancing…”

Point 12. Line 305, “These are known as…” instead of “These is known as…”

Response 12. We agree with the reviewer's suggestion. We have changed “These is known as…” to “These are known as…” in line 305.

Point 13. Line 315, “to develop” instead of “ to develope”

Response 13. We agree with the reviewer's suggestion. We have changed “ to develope” to “to develop” in line 315.

Point 14. Line 329, please correct the sentence as ”And the second one is the direct…”

Response 14. We agree with the reviewer's suggestion. We have changed the sentence in line 329 to “The second one, is the in vivo inoculation of synthetic nucleic acid-encoded T-biAb by using messenger RNA and plasmid DNA”.

Point 15. Line 342, please correct the sentence as ”…on malignant cells and that antigens is…”

Reponse 15.  We agree with the reviewer's suggestion. We have changed the sentence in line 342 to “This is possible because most hematologic neoplasms fulfill two important features for an effective biAbs therapy. The first one is that antigens from hematologic malignancies are mainly (or only) expressed on malignant cells, leading to a reduction of on-target/off cancer toxicity. The second one is that the antigen is strongly associated with the malignant phenotype, leading to a reduction of antigen loss variants”.

Point 16. Line 346, “…CD19 and CD20” instead of “ …CD19 an CD20”

Response 16. We agree with the reviewer's suggestion. We have changed “…CD19 an CD20” to “…CD19 and CD20” in line 346.

Point 17. Line 390, “…solid malignancies” instead of “…solid malignances”

Response 17. We agree with the reviewer's suggestion. We have changed “…solid malignances”  to “…solid malignancies” in line 390.

Point 18. Line 391, “…hypoxia-induced immunosuppression…” instead of “…hypoxic-induced immunosuppression…”

Response 18. We agree with the reviewer's suggestion. We have changed “…hypoxic-induced immunosuppression…”  to “…hypoxia-induced immunosuppression…” in line 391.

Point 19. Line 408, “…expresses GPC3” instead of “…express GPC3”

Response 19. We agree with the reviewer's suggestion. We have changed “…express GPC3”  to “…expresses GPC3” in line 408.

Point 20. Line 432, “improve their specificity” instead of “improve its specificity”

Response 20. We agree with the reviewer's suggestion. We have changed “improve its specificity” to “…“improve their specificity” in line 432.

Point 20. Line 439, “lead to a more …” instead of “lead a more…”

Response 20. We agree with the reviewer's suggestion. We have changed “lead a more…”  to “…“lead to a more …” in line 439.

Point 21. Line 442, please correct the sentence as “BiTAbs have been demonstrated…” 

Response 21. We agree with the reviewer’s suggestion. We have performed the suggested change.

Round 2

Reviewer 2 Report

In the revised version the authors adequately addressed the comments that were made by me and the other reviewer. Therefore, the revised manuscript can now be accepted for publication in "Pharmaceutical"